# The Acute VertEbRal AugmentaTion (AVERT) study: protocol for a randomised controlled, feasibility trial of spinal medial branch nerve block in hospitalised older patients with vertebral fragility fractures

Chia Wei Tan [1], Maribel Cameron,[1] Yuriy Arlachov,[2] Anastasios Bastounis,[3] Simon Bishop,[4] Michal Czernicki,[5] Avril Drummond [6], Apostolos Fakis,[7] Dritan Pasku,[8] Opinder Sahota [1,8]

For numbered affiliations see end of article.

**Correspondence to**
Dr Chia Wei Tan;
cwtan@doctors.org.uk

## ABSTRACT

**Introduction** Vertebral fragility fractures (VFFs) are the most common type of osteoporotic fracture found in older people, resulting in increasing morbidity and excess mortality. These fractures can cause significant pain, requiring admission to hospital. Vertebroplasty (VP) is effective in reducing pain and allowing early mobilisation in hospitalised patients. However, it may be associated with complications such as cement leakage, infection, bleeding at the injection site and fracture of adjacent vertebrae. It is also costly and not readily accessible in many UK hospitals.

A recent retrospective study reported that spinal medial branch nerve block (MBNB), typically used to treat facet arthropathy, had similar efficacy in terms of pain relief compared with VP for the treatment of painful VFF. However, to date, no study has prospectively compared MBNB to VP. We therefore propose a prospective feasibility randomised controlled trial (RCT) to compare the role of MBNB to VP, in hospitalised older patients.

**Method** A parallel, two-arm RCT with participants allocated on a 1:1 ratio to either standard care-VP or MBNB in hospitalised patients aged over 70 with acute osteoporotic vertebral fractures. Follow-up will be at weeks 1, 4 and 8 post intervention. The primary objective is to determine the feasibility and design of a future trial, including specific outcomes of recruitment, adherence to randomisation and safety. Embedded within the trial will be a health economic evaluation to understand resource utilisation and implications of the intervention and a qualitative study of the experiences and insights of trial participants and clinicians. Secondary outcomes will include pain scores, analgesia requirements, resource use and quality of life data.

**Ethics and dissemination** Ethical approval was granted by the Yorkshire & the Humber Research Ethics Committee (*reference 21/YH/0065*). AVERT (Acute VertEbRal AugmentaTion) has received approval by the Health Research Authority (*reference IRAS 293210*) and is sponsored by Nottingham University Hospitals NHS Trust

### Strengths and limitations of this study

⇒ The two interventions are proven, safe, surgical interventions, already used in existing healthcare practice and adverse events from both interventions are already well documented.
⇒ The descriptive analysis on effectiveness of outcome measures will inform hypothesis testing in a future definitive trial, including levels of variability required to determine power.
⇒ It will determine the feasibility of collecting economic measures, including detailed resource use and quality of life data within the two arms, which will aid the design of a comprehensive economic evaluation in a future definitive trial.
⇒ The nested semistructured qualitative interviews will provide valuable data to inform future trial acceptability for participants and the healthcare professionals involved in their clinical care.

(*reference 21HC001*). Recruitment is ongoing. Results will be presented at relevant conferences and submitted to appropriate journals for publication on completion.
**Trial registration number** ISRCTN18334053.

## INTRODUCTION

Vertebral fragility fractures (VFFs) are the most common osteoporotic fracture, leading to both acute and chronic back pain, substantial spinal deformity, functional disability, decreased quality of life and increased mortality.[1 2] The presence of one VFF increases the risk of a new VFF fivefold, and up to 12-fold in the presence of two or more VFFs.[3 4] These fractures frequently occur with very little trauma from day-to-day activities, such as bending forward, twisting, lifting light

objects and sitting from a standing position onto a low chair.[5]

Patients with VFF may have mild to moderate symptoms; however, a significant proportion require hospitalisation due to disabling pain.[6] The incidence ranges from 10 to 20 per 10 000/year, rising to 50 per 10 000/year in those aged 80 years and over.[7] Hospitalised patients with VFF tend to be older, frail with coexisting comorbidities and have cognitive impairment. These patients have longer hospital stay and higher mortality.[7 8] Conservative (non-surgical) treatment for these patients consists of bed-rest, analgesia and, controversially, thoracolumbar bracing.[9] These are poorly tolerated, and immobilisation leads to muscle wasting, impaired rehabilitation, further bone loss and medical complications such as chest infections, deep venous thrombosis and pulmonary embolism.[10] Potent morphine-based analgesia leads to nausea, constipation and particularly in those with cognitive impairment and delirium.[11] Moreover, one in five patients is readmitted within 30 days.[12]

Vertebroplasty (VP) is a minimally invasive, image-guided key-hole procedure that involves injection of radio-opaque bone cement into the fractured vertebral body, in an effort to provide pain relief and stability.[13] Recent guidelines recommend VP as the first-line treatment for VFF, given its efficacy, cost and minimally invasive nature compared with balloon kyphoplasty (BKP), with BKP reserved for more traumatic fractures in younger people.[14] Although a Cochrane review concluded that VP had no important benefits,[15] the limitations of the review were recognised.[16 17] VP performed within 3 weeks of fracture in older hospitalised patients with severe pain has been recommended,[18 19] demonstrating a reduction in pain, pain-related disability and hospital length of stay compared with conservative medical treatment.[12 20–22] A reduction in hospital and long-term mortality has also been reported when compared with conservative treatment.[23–26]

With ageing demographics, we anticipate an increasing demand for VP in hospitalised patients nationally. From our local audit data, 81 patients had VP in Nottingham last year. This amounted to a cost of £650 000, with a single level VP costing £4500. The literature reports complications (range 2.2%–3.9%) related to VP, which includes cement leakage, infection, bleeding at the injection site, fracture of the ribs, posterior elements or pedicle.[26]

Recent studies have shown that steroid injections at the facet joint to block the medical branches from the dorsal rami spinal nerve roots—medial branch nerve block (MBNB)—may be effective in managing pain in patients with VFF.[27] A retrospective study demonstrated that MBNB had similar efficacy in terms of pain relief and radiological changes after 2 years of follow-up[28]; however, its application in older hospitalised patients needs to be explored. We hypothesise that MBNB will be as effective clinically and cost-effective compared with VP, will be more readily accessible via anaesthetic and interventional radiology service, and will eliminate the risks of adjacent vertebral fracture and cement leakage associated with VP.

## METHODS AND ANALYSIS
### Aims
The ultimate aim of the feasibility trial is to prepare the foundations to design and conduct a multicentre, randomised controlled trial (RCT) of hospitalised older people with painful VFF in order to evaluate the clinical and cost-effectiveness of MBNB compared with routine surgical treatment.

### Objectives
The feasibility of conducting a definitive RCT will be determined by considering the following objectives:
► Determine the number of patients who meet the eligibility criteria in addition to the recruitment (including willingness to be randomised) and retention rates of those eligible patients.
► Test several outcome measures for assessment of mobility, pain and quality of life, for their potential use as a primary outcome measure.
► Calculate means and SD for the quantitative measures to allow hypothesis testing and development of the analysis plan for a future definitive trial.
► Evaluate ease of access and availability of information from primary and secondary care databases to determine the most efficient way of measuring patient level healthcare costs.
► Use of a qualitative nested interview study to assess participants' and clinicians' views on trial acceptability, trial processes and define the 'non-inferiority' margin to inform the design and conduct of a future definitive trial.

### Study design
The study design is a parallel, two-arm randomised controlled feasibility trial with participants individually randomised on a 1:1 ratio to continue with their planned surgical care-VP or MBNB treatment. Embedded within the feasibility study will be a health economic analysis to understand resource utilisation and implications of the intervention and a qualitative study which will focus on the experiences of participants and clinicians involved in the study, their insights and recommendations for improving trial acceptability and processes.

### Participants
This is single site study (Queens Medical Centre, Nottingham University Hospitals NHS trust) with a catchment population of 800 000. Participants presenting with acute painful VFF and awaiting spinal surgery will be approached for recruitment into the study.

The inclusion and exclusion criteria are as follows:

### Inclusion criteria
► Patients aged 70 years and over admitted to hospital.
► Ambulatory prior to injury.

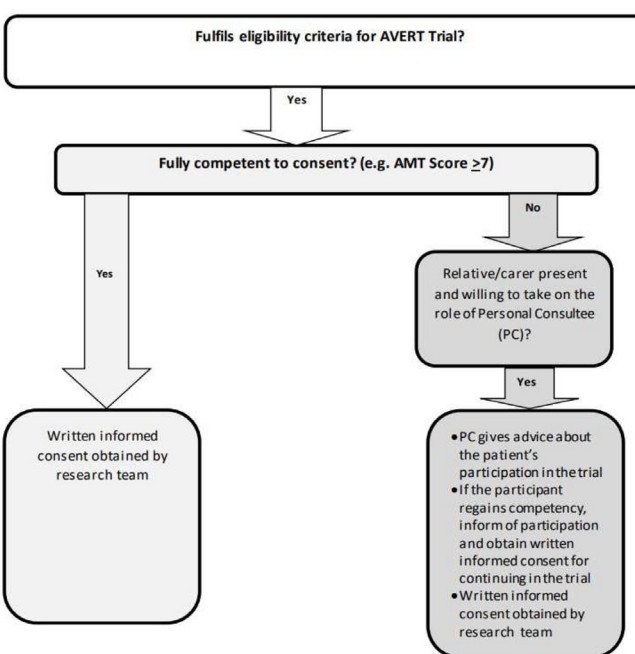

**Figure 1** Flowchart illustrating consent process for the avert trial. AMT, Abbreviated Mental Test; PC, personal consultee.

- ► <6 weeks from date of injury.
- ► Numeric Rating Scale (NRS) pain score 7 or more on standing and not responding to analgesia.
- ► MRI confirmed oedema at the site of fracture.

### Exclusion criteria
- ► Debilitating chronic back pain not relieved despite opiate use.
- ► Substantial fracture retropulsion; acute infection or spinal malignancy.
- ► Three or more acute vertebral fractures.
- ► Receiving palliative care.
- ► Spinal deformity which contraindicates VP.

### Study procedures
#### Recruitment
Patients who fulfil the eligibility criteria will be invited to participate. Those who indicate an interest will be introduced to the research team and details of the study will be explained: a patient information sheet (PIS) will be provided. Patients will have 24 hours to consider participation before providing informed consent. Consent (online supplemental file) will be obtained in accordance with Good Clinical Practice guidance and will include consent for potential inclusion in the qualitative interview study. Figure 1 illustrates the process of obtaining consent for the study.

An Abbreviated Mental Test (AMT) on admission for potential participants will be used as a screening tool for capacity assessment. A patient is deemed to have capacity if their AMT is ≥7/10 at either point of assessment. If AMT is documented as <7/10, then it will be repeated by the research team at the time of screening, and prompt a capacity assessment in relation to research.

For potential participants who are unable to provide consent, relatives or carers will be approached in the capacity of the participants' personal consultee.

Patients who decline to take part, or consultees who decline the patient's participation in the study, will be asked if they would be willing to share their reasons. The information will be valuable for us to improve future trial design and acceptability. However, there will be no requirement to do so and it will not affect the care that their relative/friend receives in hospital if they decline to share this information. The findings will be tabulated into the final results.

### Randomisation
Participants will be randomly allocated to routine care VP or MBNB. Allocation (block randomisation) will be in a 1:1 ratio via a secure web-based system (RedCap Cloud). VP and MBNB will be undertaken within 72 hours of randomisation where possible. Participants and their General Practitioners (GPs) will be notified of their allocated arm in the study and a record of randomisation will be made in the participant's medical notes.

Due to the nature of the study, it will not be possible to blind the participant or staff to VP surgery or MBNB.

### Intervention
#### Intervention group
This will be performed by a consultant anaesthetist or an interventional radiologist trained in MBNB.

Bilateral MBNB will be performed targeting facet joints above and below the vertebral fracture. Fluoroscopy will be used to assess the optimal position of the needle. A mixture of 0.5% bupivacaine with 40 mg depomedrone will be used. Each medial branch will be blocked with 1–1.5 mL solution.

### Standard care group
#### Vertebroplasty
This will be performed by an interventional radiologist or spinal surgeon under general anaesthesia or conscious sedation using fluoroscopy. The approach will be bipedicular or unipedicular, aiming the placement of the Jamshidi needle (8–12 G in size) at the centre of the vertebral body. A bony biopsy will always precede the injection of the polymethylmethacrylate bone cement. Approximately, a volume of 2–5 mL of the cement will be injected in the vertebral body according to the level and the morphology of the fracture, trying to minimise any leakage.

Usual postoperative care and monitoring will follow after the surgery. Participants will be encouraged to mobilise as pain allows and be prescribed analgesia as required.

### Outcomes
#### Feasibility study outcomes
- ► Number of eligible patients.
- ► Rate of participant recruitment and randomisation.

- ► Reasons why participants are not recruited or randomised.
- ► Rate of participant adherence to randomisation (cross-over) and retention.
- ► Completion of study rates and reasons for non-completion.
- ► Completeness of data.
- ► SD and effect size of potential outcomes for subsequent definitive trial.
- ► Time from randomisation to delivery of the intervention.

### Assessment of outcome measures used in feasibility study for its potential use in a future definitive trial

- ► Functional disability as measured by the 24-point Roland Morris Disability Questionnaire (RMDQ)[29] (this is the most commonly used outcome measure in previous studies).
- ► Pain as measured by the 0–10 Numeric Pain Rating Scale (NRS-11).[30]
- ► Quality of Life as measured by the EuroQol (EQ5D-5L)[31] and where appropriate, proxy EQ5D-5L.[31]
- ► Activities of daily living as measured by the Nottingham Extended Activities of Daily Living (NEADL) scale.[32]
- ► Abbreviated Mental Test (AMT) for cognition assessment.[33]
- ► Montreal Cognitive Assessment for cognition assessment.[34]
- ► Charlson Comorbidity Index as a predictive tool for 1-year mortality.[35]
- ► Clinical Frailty Scale (CFS)—frailty assessment.[36]
- ► Sociodemography data including age, sex, socioeconomic status.
- ► Record of pain medication use (using the opioid dose equivalence table).

### Analgesia requirement

Analgesia requirement will be recorded as follows: each medication will be classified as a strong opioid (including oxycodone, morphine, fentanyl, pethidine, hydromorphone, buprenorphine and tramadol), mild opioid (including medications containing codeine or dextropropoxyphene) or non-opioid medication (including paracetamol and non-steroidal anti-inflammatory drugs). The participant will be given a score of 0, 1 or 2 in each of these three categories depending on the number of concurrent different medications being taken within each category. Opioid medication will also include a calculation of the oral Morphine Equivalent Daily Dose using the Opioid Dose Equivalence score.[37]

### Follow-up

Participant flow through the study is summarised in figure 2. Follow-up timings will be counted from the time of intervention. Participants will be followed up face to face at week 1 (±3 days) (while in hospital), week 4 (±7 days, telephone interviews) and week 8 (±7 days, telephone interviews).

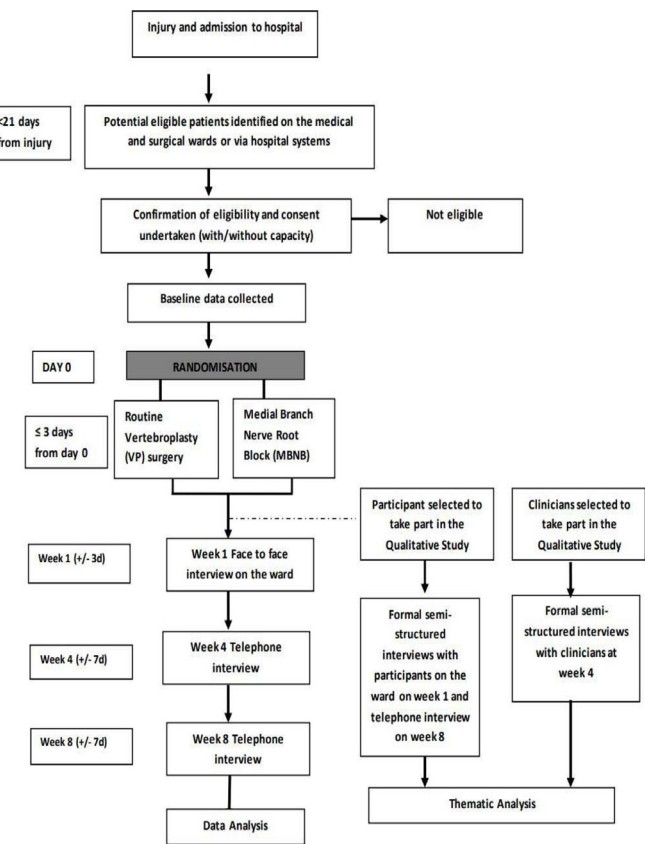

**Figure 2** Participant flow through the trial including the timings of data collection.

Follow-up outcome measures will include the following:
- ► Participant still living (established by the hospital's NHS spine portal enquiry).
- ► Hospital length-of-stay (ascertained by the hospital electronic database, supplemented by review of medical notes by a different member of staff to maintain blinding, if necessary).
- ► Unplanned hospital readmission within 28 and 91 days post discharge (ascertained by the hospital electronic database, supplemented by review of medical notes if necessary).
- ► Health economics/resource utilisation (patient and NHS costs).
- ► Time from randomisation to delivery of the intervention.
- ► Quality of life measures using RMDQ,[29] EQ-5D-5L.[31]
- ► NEADL.[32]
- ► NRS.[30]

### Qualitative assessments

The study will be complemented by a nested semistructured interview to provide essential insights into the feasibility, design and conduct of a definitive trial. This will focus on the experiences of participants (n=10) and clinicians (n=5) in the study, their insights and their recommendations for improving trial acceptability and processes.

A range of participants with different characteristics which we believe might lead them to have a different opinion or experience will be invited for the interview. For example, participants of both genders; participants from both arms of the study, a spectrum of participants on the clinical frailty scale; some patients who may have particular communication needs (eg, deafness, English as a second language speakers). Personal consultees of participants will also be invited to participate in the interview. Similarly, we wish to explore clinical staffs' thoughts about participant recruitment (eligibility and randomisation), an acceptable non-inferiority margin, as well as reflect on the process of integrating the research with the clinical service. Clinical staff will be identified by the study lead, who will distribute an invitation letter/email and information sheet to eligible staff. All clinicians will be provided with a participant information leaflet, and asked to give written, informed consent, using a dedicated clinician interview consent form.

All interviews will be audio recorded, transcribed in full and anonymised using a professional transcription service.

### Sample size calculation

Approximately 200 patients present with acute VFF each year (local audit data) of which approximately one-third require VP. We propose to recruit for 10 months, in which time we expect to approach approximately 50 patients listed for VP. Assuming 80% are eligible, and with a 70% consent rate, we expect to recruit 30 participants. Allowing for a 10% 2-month attrition rate, 28 participants will complete the study. By recruiting 30 patients, we will be able to estimate a recruitment rate of 82% (CI 74% to 92%) and a retention rate of 90% (CI 81% to 99%). Completed follow-up on 28 participants will allow the Roland Morris Disability mean score to be estimated, with an approximate SE of 0.98, assuming an SD of 6, thus enabling future power calculations to be calculated. Sample sizes between 24 and 40 are recommended for a feasibility study,[38–40] and thus by recruiting 30 participants, we are confident we can determine the feasibility of conducting a definitive trial.

### Data analysis

A statistical analysis plan will be finalised and approved prior to database lock and commencement of data analysis. Feasibility outcomes will be summarised using appropriate descriptive statistics; mean (95% CI) for continuous and frequency (%) for categorical. Completeness and descriptive summaries of outcome data at each follow-up time point will be presented. Descriptive summaries of NHS costs (using standard unit costs) data at each follow-up time point will be presented.

### Economic evaluation

In addition, at week 8, to inform the definitive economic analysis, we will assess resource use between VP and MBNB treatments; the ease of access to information about resource use from routine database systems; and the feasibility of collecting such data.

Surgical VP and MBNB treatment resource use will be collected from the medical notes and through discussions with participants by the research staff. The cost associated with the surgery will be based on the recorded resource use for the surgery (eg, consumables, equipment, grade and number of nursing staff present during the operation). Further health resource information will be extracted from the hospital electronic system (NotIS) and the GP electronic systems by the research assistant. This will include any outpatient appointment, outpatient procedures, emergency department visits, inpatient admissions related to the study or GP visits captured at baseline, week 4 and week 8.

The unit costs of health resources will be based on national tariffs such as the Unit Costs of Health and Social Care for primary care resources, NHS Reference Costs for secondary care resources and the BNF for prescriptions. These tariffs will be estimated at 2021 prices. We will also use national databases such as the NHS Healthcare Resource Group (HRG) Tariff,[41] the Personal Social Services Research Unit (PSSRU) Costs of Health,[42] the Office of National Statistics (ONS) Bulletin Annual Survey of Hours and Earnings[43] and NHS Reference Costs (https://www.gov.uk/government/collections/nhs-reference-costs). Any unit cost that is not available will be estimated in consultation with the hospital finance department.

### Health economic analysis

We will rehearse the cost-effective analysis to inform the study hypothesis and the analysis plan for the definitive trial. The within-trial economic evaluation will determine the cost and outcome of VP and MBNB treatment from a NHS perspective. The evaluation will follow the reference case guidance for technology appraisals as set out by the National Institute for Health and Care Excellence (NICE).[44] Effectiveness will be captured using quality-adjusted life-years (QALYs) as assessed by the EQ-5D-5L[31] at 8 weeks. The primary health economic outcome of the evaluation will be the incremental cost per additional QALY gained from surgical VP and MBNB. To control for the impact of uncertainty, one-way and two-way sensitivity analyses will be performed on (but not exclusively) age, gender and baseline scores. The impact of parameter uncertainty will also be addressed using a probabilistic sensitivity analysis, allowing the calculation of 95% CIs for the incremental cost-effectiveness ratio and cost-effectiveness acceptability curves.

### Qualitative analysis

Data will be handled using the NVivo software package and analysed using a framework thematic approach. The framework will be informed by the literature around the challenges of clinical trial methodology,[38 40 45–47] and initial themes are likely to include elements such as randomisation, outcome measures, communication and

feedback. Data from each interview will be mapped to the thematic tables.

## Harms

The intervention is not testing a new surgical technique or medical treatment, and VP and MBNB are recognised as safe and routinely used procedures, hence adverse events (AEs) will be recorded in the CRFs, but will not require expedited reporting to the sponsor or REC.

All AEs will be reviewed by the chief investigator (CI) and recorded as part of the study outcome measures with an assessment of the severity, relation and expectation. All deaths occurring up to the final study visit and serious AEs, other than expected surgical complications, will be recorded on the Sponsor SAE form and emailed to the sponsor within 3 days of a researcher becoming aware of the event. Those related to the study and unexpected will be reported to the REC within 15 days. Events will be followed up until resolved or a final outcome has been reached.

## ETHICS AND DISSEMNATION
### Patient and public involvement

The study's patient and public involvement (PPI) group are members of the Royal Osteoporosis Society's Nottingham Support group. The topic is well recognised among the PPI who highlighted this area for further research. The member's experiences have greatly influenced the design of the research study and the choice of proposed outcome measures of the study. They will be able to comment on aspects of the study design and contribute to production of questionnaire booklets, information sheets and other documents to ensure these are understandable and acceptable to patients. They will also be invited to comment on methods of sharing the study findings and support the writing of the definitive, future trial application.

### Dissemination policy

The team's dissemination strategy aims to target relevant policy makers and patient groups through our PPI representative. We will publish the study protocol and the full report of AVERT will be available on the NIHR RfPB website. Results from the trial will also be submitted for presentation at scientific meetings and conferences targeted at clinicians working with older people, trauma and spinal surgery.

### Study registration and approvals

AVERT has received approval from the Research Ethics Committee (REC—Yorkshire & The Humber—Bradford Leeds, reference number 21/YH/0065), Health Research Authority (HRA) and the Nottingham Queens Medical Centre Research & Innovation department. Nottingham University Hospitals NHS Trust will act as sponsor to this study. The study has been registered on a clinical trials database (https://www.isrctn.com, reference number ISRCTN 18334053, pre-results).

## DISCUSSION

Across Europe, 22 million women and 5 million men have osteoporosis, with 4 million new fragility fractures occurring a year, at an estimated health and social care cost of £37 billion[48] which is predicted to double by 2050.[49] One in 2 women and 1 in 5 men over the age of 50 will experience and osteoporotic fracture in their lifetime, with this rising by 25% over the next 5 years.[50] In the UK, around 3.5 million people have osteoporosis, with an annual incidence of 500 000 of new fragility fractures.[51] VFFs are the most common osteoporotic fracture. Prevalence studies suggest that 20% of women aged over 80 years have sustained a VFF.[52]

The majority of those with pain symptoms will have mild to moderate pain; however, a significant proportion have severe pain and require admission to hospital. The annual incidence of hospitalisation with symptomatic VFF is rising (50 per 10 000/year in those aged 80 years and over),[5] and these figures could be still be underestimated as up to 70% of vertebral fractures remain undiagnosed.[5 53] In one study, it was reported that 34% of patients with VFF required acute hospital admission,[54] with an average length of stay of 15 days in the UK.[55] Hospitalised patients with VFF tend to be older, frail with coexisting comorbidities, cognitive impairment and have worse physical-health-related quality of life.[56] With the increasing numbers of VFF and more patients requiring acute hospitalisation, the healthcare resource burden within this group of patients is alarming.

Current NICE guidance on management of VFF emphasises a trial of pain optimisation before surgical intervention (vertebral augmentation) is considered. Potent morphine-based analgesia, which is frequently required, leads to nausea, constipation and particularly in those with cognitive impairment, delirium.[11] Vertebral augmentation is a general term for several techniques used to treat painful VFF, with the aim of consolidating the fracture and, when possible, achieve height restoration. VP is a minimally invasive, image-guided key-hole procedure that involves injection of radio-opaque bone cement into the fractured vertebral body, in an effort to provide pain relief and stability.[13] Percutaneous balloon kyphoplasty (PBK) attempts to restore vertebral body height by inflating a balloon prior to bone cement injection.[57] Although the recent Cochrane review has debated the effectiveness of VP,[15–17] limitations in the review are recognised.[16 17] Emerging evidence suggests that the optimal benefits are seen when VP is performed within 3 weeks of fracture in hospitalised older patients[19 20]

Debate has been generated regarding the effectiveness of VP and its application in patients with vertebral fractures following a Cochrane review. However, the data are limited for VP and its application for inpatients,[18 19] even less so for older hospitalised patients. The VAPOUR trial,

which had a subgroup of inpatients (59%) with vertebral fractures, demonstrated that VP showed benefit in terms of pain reduction mortality, length of stay in hospital.[19] The other trials (VERTOS4,[58] Kalllmes *et al*,[59] Buchbinder *et al*[60]) highlighted in the Cochrane review had no inpatients and recruited from an outpatient setting. As far as we are aware, the AVERT trial will be the first trial to focus exclusively on hospitalised older people.

It is also worth noting that previous trials did not include patients who had a diagnosis of dementia or delirium during recruitment. As cognitively impaired older patients presenting to hospital with VFF account for a large proportion of the real-world VFF cohort,[61] it would severely affect the validity and generalisability of the trial to exclude them. However, we are unaware of any published RCT on VP that have included patients with cognitive impairment. We believe that this cohort of patients would benefit most from early intervention for their painful VFF as they may struggle to comply with regular analgesia intake, and opiate-based analgesia which is usually required may have subsequent side effects that will precipitate delirium itself.

Additionally, our participants will receive treatment for their painful VFF with either VP, a NICE approved treatment for VFF, as our control group, or our trial intervention with MBNB. Should MBNB prove as effective as VP in controlling pain, it also negates the need for a general anaesthetic for this population group, which is viewed as potentially more harmful in the frail, older population. In this trial, we also propose that surgical management should be considered earlier in the treatment of hospitalisation, improving pain with the aim of preventing pain-related immobilisation and disability, therefore reducing risks of opiate-induced side effects and consequences of prolonged immobilisation. The timely intervention (within 72 hours of randomisation) we propose in this feasibility trial will be dependent on other factors such as timely identification of acute VFF, MRI imaging availability, operator and theatre availability, where more emergent procedures will take precedence. This is in addition to the already growing emergency list from the impact of the COVID-19 pandemic.[62 63]

The alternative solution to this is to devise a new clinical pathway for these patients. Hence, it is essential that the feasibility trial is delivered within the framework of the existing healthcare service in order to facilitate implementation. As we may be potentially expanding clinical workload in other departments (ie, interventional radiology and anaesthesia) should MBNB prove effective, the feasibility of this must be explored and limitations and constraints in an already pressured healthcare system. Cost-effectiveness of MBNB must also be explored compared with standard care VP, hence, the inclusion of an economic evaluation. A benefit of a new clinical service would be that MBNB could be delivered in local hospitals via anaesthetic or interventional radiology departments. This facilitates early treatment

for painful VFF and eliminates the need for our patients to travel to regional spine centres for VP. This sentiment was echoed in a UK study which highlighted concerns regarding access to VP nationally for our older hospitalised patients with VFF.[64]

The key outcomes address questions posed by the possible limitations of conducting such a trial within an existing public health service, specifically to recruitment and adherence to randomisation. We anticipate that this feasibility study will provide information and data to test our hypothesis that MBNB is as cost-effective and safe compared with VP for the treatment of acute VFF in hospitalised older patients. With national organisational leads pushing for better outcomes for frail older patients,[65 66] we hope that this trial will be a further catalyst for change nationally in the management of older hospitalised patients with VFF.

## Trial status

The trial is currently open for recruitment, with 12 patients currently enrolled (November 2021).

We aim to complete the study in September 2022.

**Author affiliations**
[1]Health Care of the Older People, Nottingham University Hospitals NHS Trust, Nottingham, UK
[2]Department of Radiology, Nottingham University Hospitals NHS Trust, Nottingham, UK
[3]Division of Epidemiology & Public Health, University of Nottingham, Nottingham, UK
[4]Nottingham University Business School, Nottingham University, Nottingham, UK
[5]Department of Anaesthesia, Nottingham University Hospitals NHS Trust, Nottingham, UK
[6]School of Health Sciences, University of Nottingham, Nottingham, UK
[7]Derby Clinical Trials Support Unit, Royal Derby Hospital, Derby, UK
[8]Centre for Spinal Studies and Surgery, Nottingham University Hospitals NHS Trust, Nottingham, UK

**Acknowledgements** We would like to express our sincere thanks and gratitude to our PPI representative (LW) for her contribution to the original protocol and the NIHR RfPB for funding this study.

**Contributors** CWT wrote the manuscript. YA, AB, SB, MC, AD, DP, LW and OS are all key protocol contributors providing their expertise on specific aspects of the study. OS is the chief investigator of the study. MC is the trial coordinator. AF is the lead trial statistician. All authors have read, contributed amendments to and approved the final manuscript.

**Funding** This paper presents independent research funded by the National Institute for Health Research (NIHR) under its Research for Patient Benefit (RfPB) Programme (grant reference number NIHR RfPB-201937).

**Competing interests** None declared.

**Patient and public involvement** Patients and/or the public were involved in the design, or conduct, or reporting or dissemination plans of this research. Refer to the Methods section for further details.

**Patient consent for publication** Not applicable.

**Provenance and peer review** Not commissioned; externally peer reviewed.

terminology, drug names and drug dosages), and is not responsible for any error and/or omissions arising from translation and adaptation or otherwise.

**ORCID iDs**
Chia Wei Tan http://orcid.org/0000-0001-7333-5504
Avril Drummond http://orcid.org/0000-0003-1220-8354
Opinder Sahota http://orcid.org/0000-0003-0055-7637

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
