## [Reviewer comments · BMJ Open]

ARTICLE DETAILS

TITLE (PROVISIONAL)	The Acute VertEbRal AugmentaTion (AVERT) Study: Protocol for a randomised controlled, feasibility trial of spinal medial branch nerve block in hospitalised older patients with vertebral fragility fractures
AUTHORS	Tan, Chia Wei; Cameron, Maribel; Arlachov, Yuriy; Bastounis, Anastasios; Bishop, Simon; Czernicki, Michal; Drummond, Avril; Fakis, Apostolos; Pasku, Dritan; Sahota, Opinder

VERSION 1 – REVIEW

REVIEWER	Hernández, Antonio Hospital Universitario de Gran Canaria Dr Negrin, Rheumatology
REVIEW RETURNED	04-Jan-2022

GENERAL COMMENTS	I acknowledge that this is the first time that I am a reviewer of a trial protocol. I find this study very interesting when addressing the short-term effectiveness of vertebroplasty compared to lumbar nerve block in a randomized trial. It is of practical interest especially in terms of health management and health organisation. Vertebroplasty is applied differently depending on the hospital and the country, possibly depending on the fact that there are experts in the technique with local recognition and not so much on the basis of the latest systematic reviews on its effectiveness. I congratulate the authors. This trial protocol is well written and clearly explained, it is difficult to suggest modifications when it is already in the patient recruitment phase, but here are my suggestions: The title to my understanding does not clearly reflect the comparison of the two techniques and is quite confusing, so it must be adjusted to the content of the study. On line 22 of page 4, MBBM must be corrected. On line 27 on page 4 it should be clearly detailed that these are acute osteoporotic vertebral fractures that are admitted due to severe pain. On line 47 of page 5, the wording of "... .Thrombosis and pulmonary" should be revised In line 32 on page 7, it should be added to the inclusion criteria that patients present severe pain that does not respond to analgesic treatment. On line 47 on page 7, I suggest adding as an exclusion other rheumatic diseases that cause spinal pain and that may interfere with pain measurements. On page 8, line 58, the methodology of lumbar block should be better explained, in particular the medication used and what to do if several vertebrae are affected. Likewise, the vertebroplasty technique should be better described, since as it is written, it
---

	seems that each hospital will perform it in the way they consider appropriate or the way in which they usually do it.
--	---

VERSION 1 – AUTHOR RESPONSE

Reviewer Report:

Reviewer: 1

Dr. Antonio Hernández, Hospital Universitario de Gran Canaria Dr Negrin

Comments to the Author:

I acknowledge that this is the first time that I am a reviewer of a trial protocol. I find this study very interesting when addressing the short-term effectiveness of vertebroplasty compared to lumbar nerve block in a randomized trial. It is of practical interest especially in terms of health management and health organisation.

Vertebroplasty is applied differently depending on the hospital and the country, possibly depending on the fact that there are experts in the technique with local recognition and not so much on the basis of the latest systematic reviews on its effectiveness.

I congratulate the authors. This trial protocol is well written and clearly explained, it is difficult to suggest modifications when it is already in the patient recruitment phase, but here are my suggestions:

1) The title to my understanding does not clearly reflect the comparison of the two techniques and is quite confusing, so it must be adjusted to the content of the study.

Author comments:

This has now been adjusted accordingly. It now reads as:

The Acute VertEbral AugmentaTion (AVERT) Study: Protocol for a randomised controlled, feasibility trial of spinal medial branch nerve block in hospitalised older patients with vertebral fragility fractures

2) On line 22 of page 4, MBBM must be corrected.

Author comments:

- This has now been corrected and reflected in the revised edition. MBBM has been corrected to MBNB.

3) On line 27 on page 4 it should be clearly detailed that these are acute osteoporotic vertebral fractures that are admitted due to severe pain.

Author comments:

- This has now been correct and reflected in the revised edition. It is now clearly worded that vertebral fractures are acute.

4) On line 47 of page 5, the wording of "... .Thrombosis and pulmonary" should be revised

Author comments:

-This has now been amended to clarify that complications of immobilisation leads to muscle wasting, impaired rehabilitation, further bone loss and medical complications such as chest infections, deep venous thrombosis and pulmonary embolism.

5) In line 32 on page 7, it should be added to the inclusion criteria that patients present severe pain that does not respond to analgesic treatment.

Author comments:

-We have acknowledged the recommendations and the inclusion criteria now reads: Numeric Rating Scale pain score 7 or more on standing and not responding to analgesia.

6) On line 47 on page 7, I suggest adding as an exclusion other rheumatic diseases that cause spinal pain and that may interfere with pain measurements.

Author comments:

We appreciate the rationale of adding this in the exclusion criteria. However, we were not able to amend this at this point of the feasibility study as had already started our recruitment process. We will be taking the recommendation forward should we proceed to a multi-centre trial.

7) On page 8, line 58, the methodology of lumbar block should be better explained, in particular the medication used and what to do if several vertebrae are affected. Likewise, the vertebroplasty technique should be better described, since as it is written, it seems that each hospital will perform it in the way they consider appropriate or the way in which they usually do it.

Author comments:

We have taken on board the reviewer comments regarding techniques of vertebroplasty, and as correctly described, surgical approaches may differ in different locations. We have therefore attempted to be as descriptive as possible in the approach taken at our site. We have also acknowledged the comments regarding multiple vertebral fractures with regards to MBNB. As the feasibility trial only recruits patients with 1 or 2 fractures (3 or more acute fractures is part of our exclusion criteria), the technique and composition of the nerve block is described.